# Gas Transfer of Metals during the Destruction of Efflorescent Sulfates from the Belovo Plant Sulfide Slag, Russia

**Svetlana Bortnikova** [1], **Natalya Abrosimova** [1], **Nataliya Yurkevich** [1,*], **Valentina Zvereva** [2], **Anna Devyatova** [1], **Olga Gaskova** [3], **Olga Saeva** [1], **Tatyana Korneeva** [1], **Olga Shuvaeva** [4], **Nadezhda Pal'chik** [3], **Valery Chernukhin** [5] and **Alexander Reutsky** [5]

1   Trfimuk Institute of Petroleum Geology and Geophysics, Siberian Branch of the Russian Academy of Sciences, Koptug ave. 3, Novosibirsk 630090, Russia; BortnikovaSB@ipgg.sbras.ru (S.B.); AbrosimovaNA@ipgg.sbras.ru (N.A.); DevyatovaAY@ipgg.sbras.ru (A.D.); SaevaOP@ipgg.sbras.ru (O.S.); KorneevaTV@ipgg.sbras.ru (T.K.)
2   Far East Geological Institute Far East Branch of the Russian Academy of Sciences, Prospekt 100-letiya, 159, Vladivostok 690022, Russia; zvereva@fegi.ru
3   Sobolev Institute of Geology and Mineralogy, Siberian Branch of the Russian Academy of Sciences, Koptug ave. 3, Novosibirsk 630090, Russia; Gaskova@igm.nsc.ru (O.G.); Nadezhda@igm.nsc.ru (N.P.)
4   Nikolaev Institute of Inorganic Chemistry, Siberian Branch of Russian Academy of Sciences, Lavrentiev ave. 3, Novosibirsk 630090, Russia; Olga@niic.nsc.ru
5   SibEnzyme Ltd., Timakova st., 2/12, Novosibirsk 630117, Russia; Valeraquest@mail.ru (V.C.); Reoutsky@mail.ru (A.R.)
*   Correspondence: YurkevichNV@ipgg.sbras.ru; Tel.: +7-(383)-363-91-94

**Abstract:** This paper demonstrates the results of experiments for the determination of the composition of gases during the dehydration of sulfates (Na-jarosite, melanterite, and chalcanthite) collected at the surface of pyrometallurgical waste heaps. The volatilization of various elements, and vapor–gas phase transport from three sulfate groups were investigated by stepwise laboratory heating at 45, 55, and 65 °C. The sample of yellow efflorescence mainly consisted of Na-jarosite, the white efflorescence contained melanterite as the major mineral, and the blue efflorescence sample consisted of chalcanthite. These all contained a few impurities up to 5 %. The highest total dissolved solids (TDS) was found in the gas condensates from melanterite (59 mg/L), followed by chalcanthite (29 mg/L) and Na-jarosite (17 mg/L). It was determined that major and trace elements in the condensate can be trapped by water vapor and can migrate with the vapor phase during the desorption and dehydration of hydrous sulfates. X-ray diffractograms showed that Na-jarosite remained stable throughout the temperature range, whilst the separation of melanterite's structural water occurred at 40 °C, and chalcanthite completely lost two water molecules at 50 °C. The gas condensates contained acetates and formates, which could be the fermentation products of bacterial communities. Some of the strains—*Micrococcaceae* sp., *Bacillus* sp., and *Microbacteriaceae* sp.—were cultivated.

**Keywords:** metallurgical waste; sulfate dehydration; gas condensates; adsorbed and structural water; vapor-gas phase elements

---

## 1. Introduction

Metal-sulfate salts play an important role in the storage and transport of acids and metals released during the weathering of mineralized rocks, coal deposits, metallic ore deposits, and mine waste [1–4]. The role of sulfates is apparent when they are dissolved by drainage streams and rainfall [5–9], while

sulfates also play a role in the retention of oxyanions (e.g., $H_2AsO_4^-$, $SO_4^{2-}$, and $SbO_3^-$), and divalent cations (Cu, Zn, and Pb) in low-pH oxidation zones [1,10–12]. Secondary mineral compositions correlate with acid mine drainage (AMD) chemistry and the overall mineralogical assemblages and morphologies [13].

A key source of information for the assessment of AMD potential is the estimation of the emissions of trace elements and sulfur gases from sulfide tailings, typically by using geochemical, mineralogical, and geophysical (electrical resistivity tomography) techniques [14–16]. It has been established that air streams over the sulfide tailings of the Komsomolsk gold extracting plant (tailings substance breathing) contain complex mixtures of sulfur-containing gases, which are capable of carrying many chemical elements. A local anomaly of extremely low resistivity (0.3–0.5 ohm·m) might be associated with a combustion center or pore solutions with high TDS at the Belovo zinc-processing plant waste heaps in the Kemerovo region, Russia. Basic major elements (Ca, Mg, K, Na, Si, and Al), metals (Cu, Zn, Pb, and Cd), and semimetals (As, Sb, and V) have been found in gas condensates in situ. To explain the patterns and features of the behavior of these elements during phase separation in solid-gas-vapor systems in detail, laboratory experiments were undertaken at three temperatures.

The investigation of sulfates and hydrated sulfate minerals plays a key role in the interpretation of the hydrochemical history of man-made toxic tailings (and rock dumps) [2,17–21]. An investigation of the phase dehydration mechanisms of sulfate–mineral mixtures is important in determining how mine waste interacts with the local environment, and in understanding the processes by which mine waste matures with time and reacts to changes in temperature and/or humidity [22,23]. The temperature of water separation from sulfate-mineral mixtures (nonetheless, predominantly monomineral) and their transformation were studied here. Latent acidity is effectively stored in these secondary sulfate minerals, and could be underestimated when monitoring the local environment during dry summer testing (1):

$$KFe_3(SO_4)_2(OH)_6 + 3H_2O = 3Fe(OH)_3(s) + K^+ + 2SO_4^{2-} + 3H^+. \qquad (1)$$
$$\text{jarosite} \qquad\qquad \text{iron hydroxide}$$

Moreover, the studied disposal site is not standard, in terms of its mine tailings (flotation material from sulfide ore processing), rather being a burning dump composed of a mixture of slag and coke dust [14]. In this case, a study of the temperature variations acquires a specific interest. Thus, two sets of systematic experiments are described: One involved elemental volatilization in mineral-gas-vapor systems, and the other the dehydration of sulfates at three temperatures.

It was established that air streams over sulfide tailings are complex mixtures of sulfur-containing gases, which are capable of carrying many chemical elements [14–16]. The mechanism of element migration in ambient conditions was proposed [16]. However, currently, we cannot explain in detail the patterns and features of the behavior of elements during phase separation in solid-gas-vapor systems.

The purpose of this work was to determine the composition of the vapor–gas mixture separated from sulfate minerals during heating.

## 2. Materials and Methods

### 2.1. Site Description

Efflorescence samples were collected from the surfaces of the Belovo pyrometallurgical waste heaps (Belovo zinc-processing plant, Kemerovo region, Russia, Figure 1A). The plant extracted Zn from a sphalerite concentrate that was obtained from barite-sulfide ores mined at the Salair ore field. The waste material of the Belovo zinc processing plant is a slag, which is a product of pyrometallurgical smelting. This plant has operated since the 1930s until the mid-1990s, when the plant ceased operation. Approximately 1 million tons of slags containing significant amounts of sulfuric acid remained in the plant area in heaps. The slag is a loose, coarse-grained material comprising silicate glass with inclusions

of K-feldspar, olivine, spinel, alloys, and sulfides [24]. The mineralogy and internal structure of the Belovo waste heaps have been described in detail elsewhere [14,24,25]. In addition, large amounts (approximately 20–25%) of fine-grained coke occur in the waste. The slag was stored in heaps of approximately 15 m in height with a flat top and a steep slope (Figure 1B). This waste was affected by spontaneous ignition of the coke dust and subsequent burning of waste in the heaps. Due to the intensive transformation of the slag under the influence of oxidizing agents, intensified by combustion, acid high-mineralized drainage (Figure 1C), and abundant efflorescences, sulfates, consisting of Fe, Cu, and Zn, have formed on the surface (Figure 1D–F).

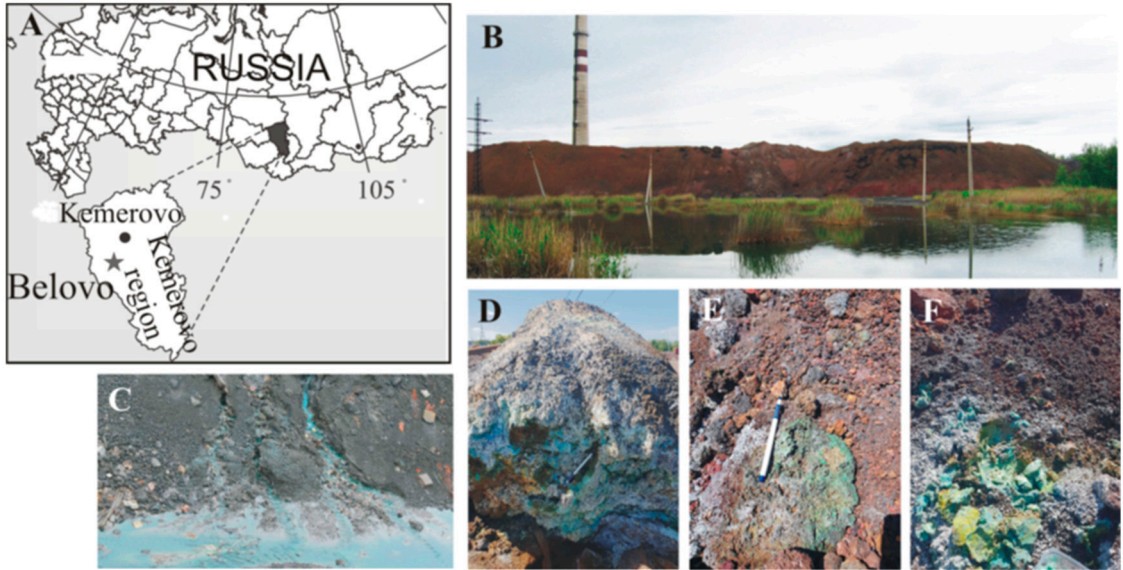

**Figure 1.** (**A**) Geographical location of the study object; (**B**) General overview of the Belovo waste heaps; (**C**) Drainage streams from inner parts of the heaps; (**D–F**) Secondary efflorescences on the surface.

### 2.2. Sample Collection and Preparation

During a field survey under hot and dry weather conditions, efflorescence samples of different colors (yellow, white, and blue) were carefully collected. The weight of each sample was approximately 0.5 kg. The samples were stored in sealed plastic containers. In the laboratory, foreign substances were removed from the samples, and then the samples were sorted by hand-picking, using a binocular microscope. The sorting was performed considering the grain color, in order to achieve monomineral samples, as far as possible.

### 2.3. Mineralogical Analyses

Powder X-ray diffractometry (XRD) was used to determine the phase compositions of the crystalline substances and their quantitative phase relationships. The XRD studies were performed on an ARL X'TRA powder diffractometer (Thermo Fisher Scientific, Ecublens, SARL, Switzerland), using CuKα radiation, a voltage of 40 kV, and a current of 25 mA. The diffraction patterns were scanned at a 2θ interval from 2° to 65°, in steps of 0.02°, and the analysis speed was 4° per minute. The morphology and composition of the individual grains were studied using a TESCAN MIRA3 LMU scanning electron microscope (Czech Republic), with an INCA Energy 450 + microanalyzer, based on the Oxford Instruments NanoAnalysis X-MAX 80 (UK) system in the laboratory of X-ray spectral analysis methods at the Sobolev Institute of Geology and Mineralogy, Novosibirsk (analyst N. S. Karmanov).

### 2.4. Condensate Collection

A condensate collection device was designed in the laboratory (Figure S1, Supplementary Material). Then, 100 g of the sample was placed in a heat-resistant beaker covered with a funnel that was connected to the bubbler inlet with a silicone hose. An air/gas mixture was pumped out of the ice-cooled bubbler through an exit port by means of a back-pressure pump (pumping speed ~2.4 L/min). The beaker was heated at a temperature interval of 45 to 65 °C on a digital magnetic stirrer, and the condensate was collected in the bubbler. After some time of heating, the condensate ceased to flow into the bubbler, indicating that the release of the structural and sorbed water had stopped. The open beaker containing the sample was then left at room temperature and humidity for 24 h. Subsequently, the vapor phase appeared again at the same heating temperature. This corresponded to the release of the newly-sorbed water, and this condensate was again collected. To collect the necessary amount of condensate (10 mL), this procedure was repeated several times. In fact, possible rehydration of the samples was observed (the evaporation and condensation water cycle).

### 2.5. Analyses of Condensates

The pH and redox potential (Eh) values of each water sample were determined in situ using a pH/°C meter (HI 9025 C, Hanna Instruments, Ronchi Di Villafranca, Italy), equipped with glass electrodes for measuring the pH (HI 1230 B, Hanna Instruments), and Pt electrodes for measuring Eh (Oxidation Reduction Potential electrode, Hanna Instruments, Woonsocket, USA). The accuracy and precision were estimated to be 5% or better. The contents of the anions ($SO_4^{2-}$, $Cl^-$, $F^-$, $NO_3^-$, and $NO_2^-$) were measured with a DIONEX ICS-2100 ion chromatograph, equipped with an IonPac AS19 (2-mm) column. Quantitative analysis was carried out using an external calibration against areas of ion peaks using Chromeleon7 software. For the replicates of multiple measurements, the relative standard deviation was at a level of ± 15% in a concentration range of 0.1 mg/L to 1000 mg/L. An Agilent 8800 ICP-MS instrument (Tokyo, Japan), equipped with a MicroMist nebulizer, was used to determine the elements in the water samples. High-purity Ar (99.95%) was used as the plasma-forming, transporting, and cooling gas. A solution of 7Li, 59Co, 89Y, and 205Tl in 2% nitric acid, with a concentration of 1 μg/L for each determined element (Tuning Solution, USA), was used for the adjustment. All measurements were conducted in three replicates (n = 3) for each element. The relative standard deviation did not exceed 13% in all measurements.

### 2.6. Control for Mineral Transformation with Heating

The temperature of the water separation from the mineral upon heating was first determined as follows. A sample was placed in a Petri dish, and was stepwise heated on an electric stove at a fixed temperature. The temperature of the sample was measured inside the dish using an LT-300 thermometer. The temperature of the water separation was fixed when a mist appeared on the Petri dish lid, then the temperature at the appearance of droplets on the lid was fixed. The phase transformations of the minerals with heating were determined by obtaining XRD diffractograms after each heating step. Each sample was heated on a WiseStir MSH-20D-Set (DAIHAN Scientific) digital magnetic stirrer at a temperature interval of 25 to 65 °C for 1 h for each 10 °C step, under ambient air conditions. After each heating step, the samples were characterized using XRD analysis to obtain a spectrum for each temperature step, showing the transformations of the mineral phases and possible dehydration.

### 2.7. Bacteria Incubation

Bacteria were incubated in the collected sulfates using an ordinary lysogeny broth medium with 1.5% agarose. For microbiological analysis, Eppendorf flasks were filled with 200 μL of the sample. Next, autoclaved distilled water was added to the flasks to a volume of 1 mL, and was thoroughly mixed for 20 minutes on a shaker. This resulted in a 50 μL slurry. Autoclaved distilled water was added to the slurry to a volume of 1 mL, and was thoroughly mixed for 10 minutes on a shaker. Thus, 1 μL of

the resulting suspension corresponded to 1/100 μL of the initial mixture. The resulting suspension was seeded onto two pairs of Petri dishes, with a value of 1 μL and 1/10 μL of the suspension per pair. The Petri dishes were incubated at 25 °C for 1 week. At the end of the experiments, the number of colonies was counted, and the taxonomic diversity of the growing strains was visually assessed. The number of strains of each group was counted for 1 μL of the suspension (1/100 μL of the original sample). Determination of the strains was performed according to standard procedures, based on morphological and biochemical methods [26].

## 3. Results

### 3.1. Characterization of Mineral Phases

Since the collected efflorescent sulfates consisted of fine-grained intergrowths, it was not possible to obtain completely monomineralic fractions. As a result, the samples for the experiments were a mixture of metal sulfates, with a predominance of one phase. The sample of yellow efflorescence (Figure 2a) basically consisted of Na-jarosite $(K,Na)Fe^{(III)}_3(SO_4)_2(OH)_6$, with a small quantity of melanterite, $Fe^{(II)}SO_4 \cdot 7H_2O$, Trace minerals of gypsum, misenite $(K_8H_6(SO_4)_7)$, and bassanite $(CaSO_4 \cdot 0.5H_2O)$ were determined (Table S1, Figure 2d).

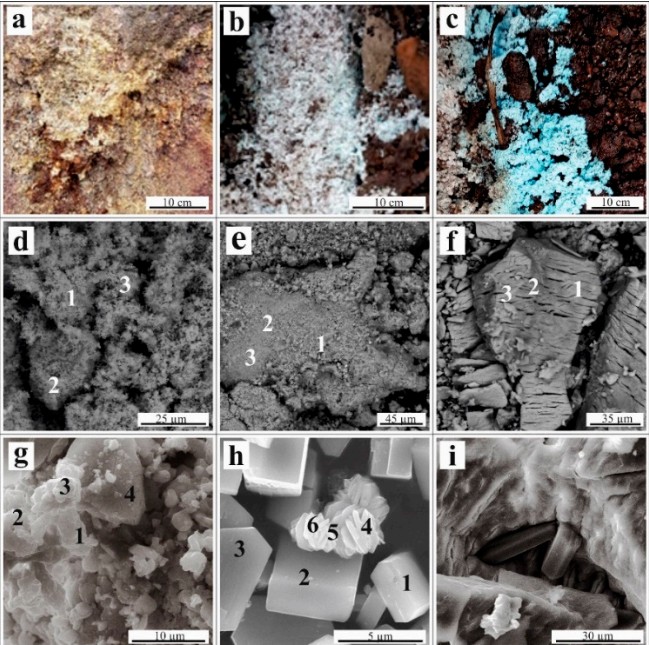

**Figure 2.** (**A–C**) Occurrence of efflorescences on the waste surface; (**D–I**) Scanning electron microscope images: (**D**)—fine-grained mass of Na-jarosite; (**E**)—melanterite with an admixture of Cu and Zn; (**F**)—chalcanthite $(CuSO_4 \cdot 5H_2O)$; (**G**)—unformed grains of namuwite $(Cu,Zn)_4SO_4(OH)_4 \cdot 4H_2O$; (**H**)—needle-like crystals of linarite $(PbCu[SO_4](OH)_2)$ associated with short-prismatic crystals of antlerite $(Cu_3(SO_4)(OH)_4)$; and (**I**)—prismatic crystals of gypsum $(CaSO_4 \cdot 2H_2O)$ associated with chalcanthite. The numbers in images (**D–I**) correspond to the locations of the energy dispersive X-ray microanalysis (Table 1).

**Table 1.** Composition of the sulfate minerals, wt.%.

| | Na-Jarosite (Figure 1D) | | | Melanterite (Figure 1E) | | | Chalcanthite (Figure 1F) | | | Namuwite (Figure 1G) | | | | Antlerite, Linarite (Figure 1H) | | | | | |
|---|---|---|---|---|---|---|---|---|---|---|---|---|---|---|---|---|---|---|---|
| | 1 | 2 | 3 | 1 | 2 | 3 | 1 | 2 | 3 | 1 | 2 | 3 | 4 | 1 | 2 | 3 | 4 | 5 | 6 |
| Fe | 32.8 | 35.63 | 33.75 | 19.65 | 18.24 | 20.48 | | | | | | | | 0.32 | 0.24 | 0.14 | 3.39 | 3.39 | 3.35 |
| Cu | 0.49 | 0.84 | 0.53 | 0.70 | 0.61 | 0.56 | 22.42 | 24.82 | 26.94 | 23.39 | 23.61 | 21.97 | 24.95 | 51.56 | 52.66 | 53.69 | 15.93 | 14.9 | 17.42 |
| Zn | 0.49 | 0.52 | | 4.23 | 1.57 | 2.19 | | | | 24.58 | 22.76 | 24.51 | 23.08 | 0.84 | 0.74 | 0.71 | | | |
| Ni | | | | | | | | | | 0.16 | 0.38 | 0.26 | | | | | | | |
| Pb | | | | | | | | | | | | | | | | | 48.1 | 51.57 | 49.35 |
| Mn | | | | | | | | | | 0.57 | 0.54 | 0.61 | | | | | | | |
| S | 13.67 | 12.16 | 13.39 | 13.42 | 12.24 | 12.63 | 10.09 | 11.31 | 12.27 | 6.40 | 7.32 | 7.26 | 9.03 | 9.2 | 9.26 | 9.29 | 9.42 | 8.41 | 8.15 |
| Si | 0.27 | | | | | | 0.10 | 0.08 | 0.20 | | | | | | | | | | |
| Al | 0.16 | 0.24 | 0.20 | 0.44 | 0.65 | 0.49 | | | | 0.31 | 0.24 | 0.24 | | 0.41 | 0.15 | 0.17 | 6.34 | 4.74 | 4.99 |
| K | 0.69 | 0.47 | 0.68 | | | | | | | | | | | | | | 0.42 | 0.33 | 0.31 |
| Na | 4.84 | 4.78 | 5.12 | | | | | | | | | | | | | | | | |
| Mg | | | | | | | | | | 1.0 | 3.12 | 2.75 | | | | | | | |
| O | 53.88 | 46.14 | 54.81 | 58.35 | 61.2 | 62.15 | 10.97 | 5.57 | 40.97 | 38.05 | 44.83 | 46.23 | 40.84 | 33.0 | 42.91 | 38.16 | 40.3 | 34.47 | 36.88 |
| Cl | | | | 0.22 | 0.30 | 0.18 | | | | | | | | | | | | | |

Melanterite was the major mineral in the white efflorescence sample (Figure 2b), and hydrous sulfates of Zn, Ni, and Ca were the minor phases, with traces of hexahydrite ($MgSO_4·6H_2O$). Since stable admixtures of Cu and Zn were present in the melanterite, the presence of Zn-melanterite ($(Zn,Cu,Fe)SO_4·7H_2O$; Table 1) microscale impurities was possible, as described by [2], citing [27].

The blue efflorescence samples (Figure 2c) consisted of a chalcanthite stoichiometric composition. Namuwite ($Zn_4(SO_4)(OH)_6·4H_2O$) and other hydrous sulfates of Cu, Ni, Ca, and Al, and hydrous arsenate cornubite ($Cu_5(AsO_4)_2(OH)_4$) were the minor phases (Table S1, Figure 2).

Bacterial strains were detected in all samples after seeding. The greatest diversity was characteristic for the medium with chalcanthite (Figure 3), in which the strains of six families were represented, the smallest for medium with Na-jarosite (only two families). However, in terms of the number of colonies, the medium obtained by adding chalcanthite was the least populated. The families, Microbacteriaceae and Micrococcaceae, were cultured only. A distinctive feature of the medium obtained with the addition of chalcanthite was the presence of the bacteria, *Staphilocuccus* sp.

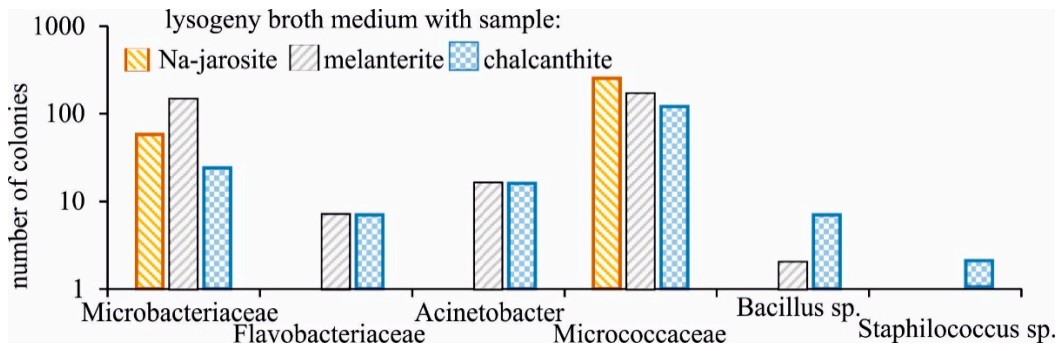

**Figure 3.** Taxonomic diversity of the growing bacteria strains on medium with the studied samples.

## 3.2. Condensate Chemical Composition

The condensates obtained by the stepwise heating of the sulfate samples were ultrafresh solutions. The pH values varied in a narrow range from 6.50 to 7.26, and the Eh was ~400 mV. This corresponds to neutral Eh/pH conditions at near ambient temperatures. The highest mineralization was determined in the condensates of the first heating at 45 °C. After this, the mineralization gradually decreased. The predominant anion in the condensates from the Na-jarosite and chalcanthite was bicarbonate, and in the condensates from the melanterite, the predominant anions were nitrates and chlorides. Ca was the major cation (7, 5, and 14 mg/L after the first heating of the jarosite, melanterite, and chalcantite, respectively); Mg was less abundant. Na and K cations made up a small amount of the samples. The highest TDS (mineralization) was found in the condensates from melanterite (59 mg/L due to the high content of $Cl^-$ and $NO_3^-$), followed by chalcanthite (29 mg/L) and Na-jarosite (17 mg/L). A feature of all the solutions was the presence of organic compounds (acetates and/or formates), which raised a question about further detailed determinations of the composition of the organic compounds in the gases.

A wide range of minor chemical elements was determined in the condensates (Table 2). Ba and Sr were the most common elements, with their concentrations reaching 330 µg/L (Ba, melanterite) and 200 µg/L (Sr, chalcanthite). This is the first direct evidence for the high migration of all alkaline earth metal ions ($Mg^{2+}$, $Ca^{2+}$, $Sr^{2+}$, and $Ba^{2+}$) in gas phases (possibly as $Me^{2+}(H_2O)_6$). The next common elements were Cu (3.4–130 µg/L), Zn (6.3–93 µg/L), aluminum (5.7–42 µg/L), and Te (6.2–48 µg/L). Mn, Pb, Ni, and Li were determined in all samples, but at lower concentrations. As was determined in the two condensates from melanterite, and Sb was found in one condensate from the Na-jarosite.

**Table 2.** The composition of gas condensates after the efflorescence samples heating, $Cl^-$–Si in mg/L, Fe–Li in μg/L.

|  | Na-Jarosite (Yellow) | | | Melanterite (Blue) | | | Chalcanthite (White) | | |
|---|---|---|---|---|---|---|---|---|---|
|  | 45 °C | 55 °C | 65 °C | 45 °C | 55 °C | 65 °C | 45 °C | 55 °C | 65 °C |
| pH | 7.22 | 7.13 | 7.25 | 7.26 | 6.50 | 7.10 | 6.56 | 6.64 | 6.83 |
| Eh | 377 | 391 | 380 | 407 | 431 | 405 | 422 | 395 | 402 |
| $Cl^-$ | 2.0 | 1.1 | 1.4 | 13 | 3.0 | 1.6 | 0.18 | <0.1 | 0.33 |
| $NO_2^-$ | 0.73 | 0.67 | <0.1 | 1.9 | <0.1 | 1.7 | <0.1 | <0.1 | <0.1 |
| $NO_3^-$ | 2.7 | 0.60 | 0.90 | 28 | 5.7 | 1.6 | 0.93 | <0.1 | 2.75 |
| $SO_4^{2-}$ | 0.53 | 0.22 | 0.27 | 9.3 | 1.1 | 0.43 | 5.6 | 16 | 2.3 |
| $HCO_3^-$ | 5.4 | 3.8 | 5.2 | <0.1 | <0.1 | <0.1 | 19 | 12 | 14 |
| Ca | 6.6 | 2.7 | 2.0 | 4.8 | 1.3 | 0.40 | 14 | 9.0 | 8.2 |
| Mg | 1.5 | 0.46 | 0.35 | 0.69 | 0.030 | 0.026 | 4.3 | 1.6 | 0.43 |
| Na | 1.4 | 0.55 | 0.38 | 0.45 | <0.1 | <0.1 | 3.4 | 1.4 | 0.37 |
| K | 0.67 | 0.53 | 0.35 | 0.21 | 0.10 | 0.099 | 0.40 | 0.34 | 0.075 |
| Si | 0.45 | 0.21 | 0.21 | 0.22 | 0.23 | 0.013 | 0.79 | 0.74 | 0.51 |
| Fe | 0.14 | 0.25 | 1.0 | 21 | 17 | 6.0 | 9.2 | 14 | 3.8 |
| Al | 7.4 | 6.5 | 5.7 | 42 | 52 | 22 | 23 | 20 | 9.5 |
| Ba | 71 | 99 | 100 | 330 | 97 | 81 | 66 | 76 | 77 |
| Sr | 77 | 32 | 25 | 190 | 24 | 9.9 | 200 | 98 | 69 |
| Mn | 1.1 | 0.66 | 0.37 | 2.8 | 1.9 | 0.48 | 1.5 | 1.5 | 0.58 |
| Cu | 3.4 | 3.4 | 3.4 | 16 | 21 | 3.2 | 130 | 77 | 110 |
| Zn | 6.3 | 8.3 | 9.5 | 36 | 52 | 6.6 | 93 | 84 | 57 |
| Pb | 2.2 | 2.2 | 2.4 | 3.5 | 4.7 | 2.5 | 9.1 | 6.2 | 25 |
| Cd | 0.68 | 0.72 | 0.84 | <1.0 | <1.0 | <1.0 | <1.0 | <1.0 | <1.0 |
| Co | 0.83 | 0.84 | 0.94 | 2.5 | 2.3 | 1.8 | <0.2 | <0.2 | <0.2 |
| Ni | 2.5 | 2.5 | 2.8 | 4.3 | 4.3 | 4.2 | 3.8 | 13 | 2.0 |
| As | <5.0 | <5.0 | <5.0 | <5.0 | 7.0 | 5.8 | <5.0 | <5.0 | <5.0 |
| Sb | <1.0 | <1.0 | 3.4 | <1.0 | <1.0 | <1.0 | <1.0 | <1.0 | <1.0 |
| Te | 6.2 | 6.6 | 7.7 | 7.7 | 7.1 | 8.7 | 16 | 48 | 13 |
| P | 13 | 26 | 39 | 40 | 18 | 85 | <10 | <10 | <10 |
| Li | 0.68 | 0.13 | 0.14 | 2.4 | 0.24 | 0.80 | 5.4 | 0.72 | 0.22 |

Note. Besides chemical elements, ion peaks for organic compounds (acetates or/and formates) were found in all condensate samples using ion chromatography, but quantitative determination analysis was impossible due to a lack of external calibration.

All solutions contained organic compounds—acetates and/or formates. Presumably, they are the waste products of the vital activities of bacterial communities. Acetate results from acetic acid fermentation, and formates—for example, sodium—are metabolites of methylotrophic bacteria. In the efflorescence samples selected to parallel those used in the above-described experiments, the following families of bacteria were cultivated: Microbacteriaceae, Flavobacteriaceae (*Flavobacterium* sp.), Moraxellaceae (*Acinetobacter* sp.), Micrococcaceae, and Bacillaceae (*Bacillus* sp.).

The jarosite was the most stable mineral when heated, with melanterite being the least stable according to the results of the experiments. Melanterite is a mineral of ferrous Fe (Fe(II)), and the Fe concentration in the gas condensate was the highest at 45 °C (i.e., 21 μg/L) compared to the jarosite (0.14 μg/L). However, the content of the major cations (Ca, Mg, Na) and some metals (Sr, Cu, Zn, Pb) was markedly higher in the condensates separated from chalcanthite (Figure 4). The concentrations of Fe, Al, Ba, Mn, and P were higher in the condensates from the melanterite. The highest K concentrations were in the condensates from the Na-jarosite, due to the decay of misenite ($K_8H_6(SO_4)_7$).

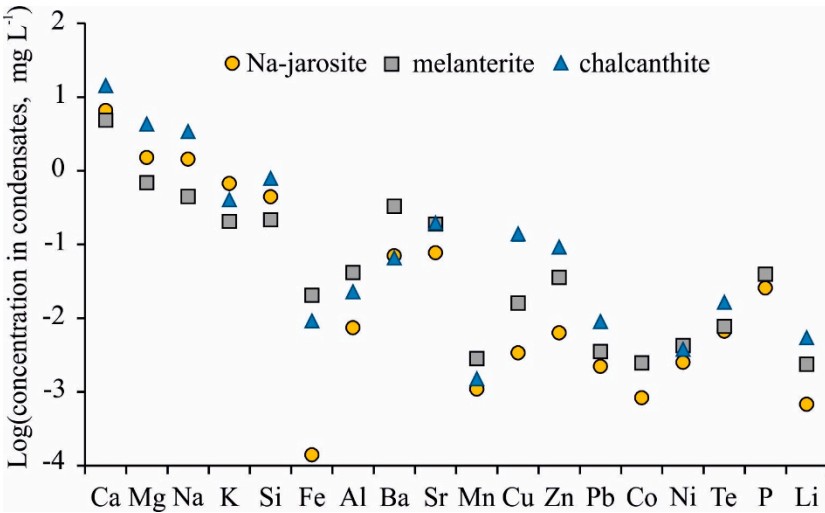

**Figure 4.** Mean element concentrations in gas condensates from Na-jarosite, melanterite, and chalcanthite.

### 3.3. Transformation of the Sulfate Minerals

It became clear that the chemical admixture in the gas condensates from the jarosite, melanterite, and chalcanthite corresponded to impurities from the contaminant minerals. Therefore, a study was performed on the main mineral transformations during dehydration with heating.

The temperature of the dehydration was the lowest for melanterite ($FeSO_4·7H_2O$; in accordance with its ultimate instability during the condensate removal). The first mist appeared on the lid of the Petri dish at 27 °C (Tm), and abundant drops appeared at 45 °C (Td). For the chalcanthite, mist on the lid formed at 40 °C, and droplets formed at 60 °C. For the Na-jarosite, water began to separate at 48 to 50 °C, and drops appeared on the lid at 55 °C (Table 3). The temperature of dehydration attributable to the water release is related to the structural position of the water.

**Table 3.** Dehydration temperature of the crystalline hydrates, °C.

| Sample | Tm | Td |
|---|---|---|
| Na-Jarosite | 48–50 | 55 |
| Melanterite | 27 | 40–45 |
| Chalcanthite | 40 | 60 |

The phase transformation during heating showed that the Na-jarosite remained stable throughout the temperature range (Figure 5), although the composition of the impurity mineral, melanterite, changed this, decreasing the intensity of the diffraction peaks. Minerals of the jarosite group are common and, according to [28–30], can be formed under a variety of natural conditions at temperatures of 25 to 110 °C. Clearly, the water separating from the Na-jarosite involved water molecules that were adsorbed on the surface of a single crystal/grain (this corresponded to the lowest content of Fe and $SO_4^{2-}$ in the gas condensate).

The data on the dehydration temperature of $7H_2O$-melanterite (27–45 °C) are in good agreement with the experimental data [9]. During the heating process, the near-infrared spectra of the minerals revealed changes in the samples at the first temperature step (30 °C), where the melanterite changed to $4H_2O$-rozenite [9]. In our case, the intensity of the diffraction peaks of melanterite appreciably decreased at a temperature of 40 °C, and the melanterite was partly transformed into $5H_2O$-siderotil. The other XRD peaks represented the contaminant mineral peaks—retgersite ($NiSO_4·6H_2O$) and gunningite (($Zn$,$Mn^{2+}$)$SO_4·H_2O$), hexahydrite, and gypsum (Figure 6). These caused the presence of the corresponding elements in the gas condensates.

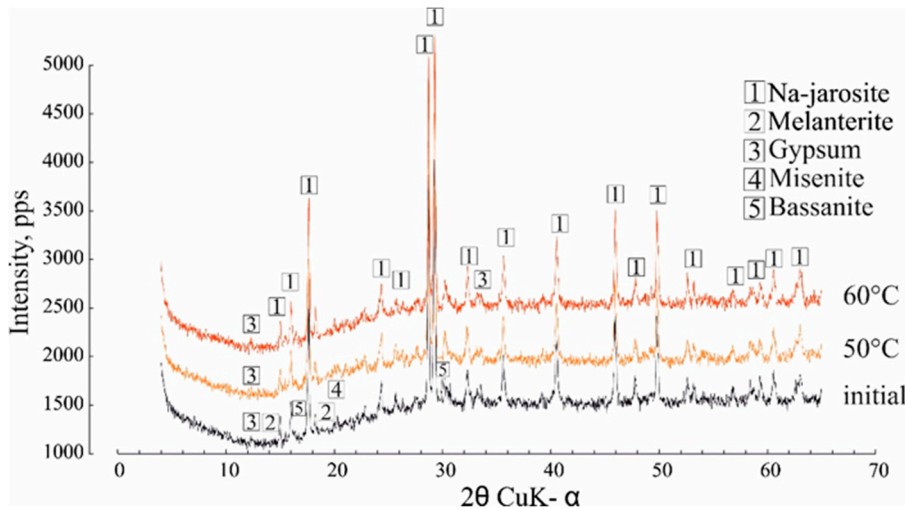

**Figure 5.** The effect of heating at 50 °C and 60 °C on the Na-jarosite XRD characteristics.

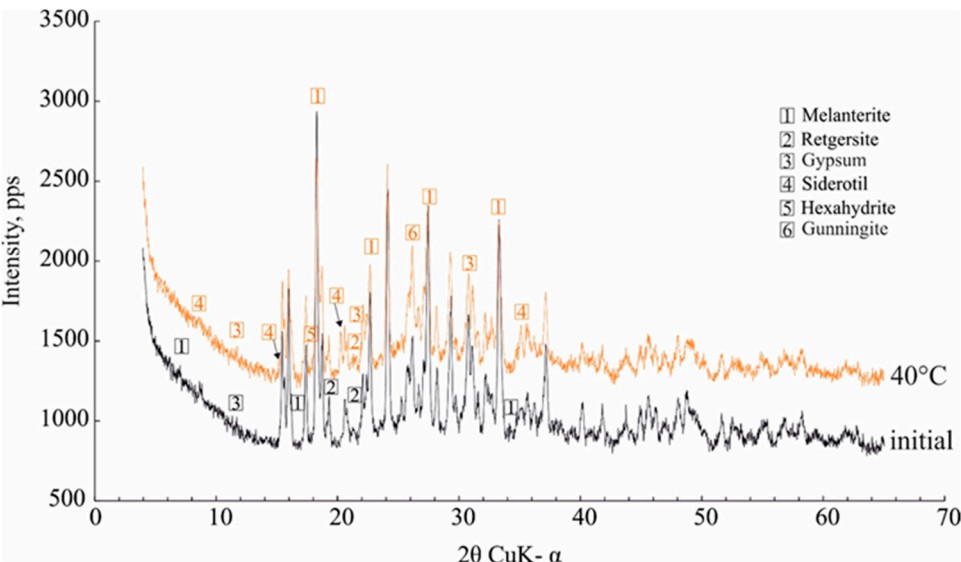

**Figure 6.** The effect of heating at 40 °C on the melanterite XRD characteristics.

The dehydration of $5H_2O$-chalcanthite began at 40 °C, when the first lines of trihydrate ($CuSO_4 \cdot 3H_2O$) bonattite appeared; the chalcanthite completely lost $2H_2O$ molecules and transformed to bonattite at 50 °C (Figure 7). None of the transformations were as vividly manifested in the XRD patterns. The gypsum remained unchanged. At first glance, it was observed that the powder diffractogram of the predominant chalcanthite was the most complex, with six accurately identified minerals. This is why the major content of the cations and metals (Sr, Cu, Zn, Pb) was markedly higher in the condensates separated from chalcanthite.

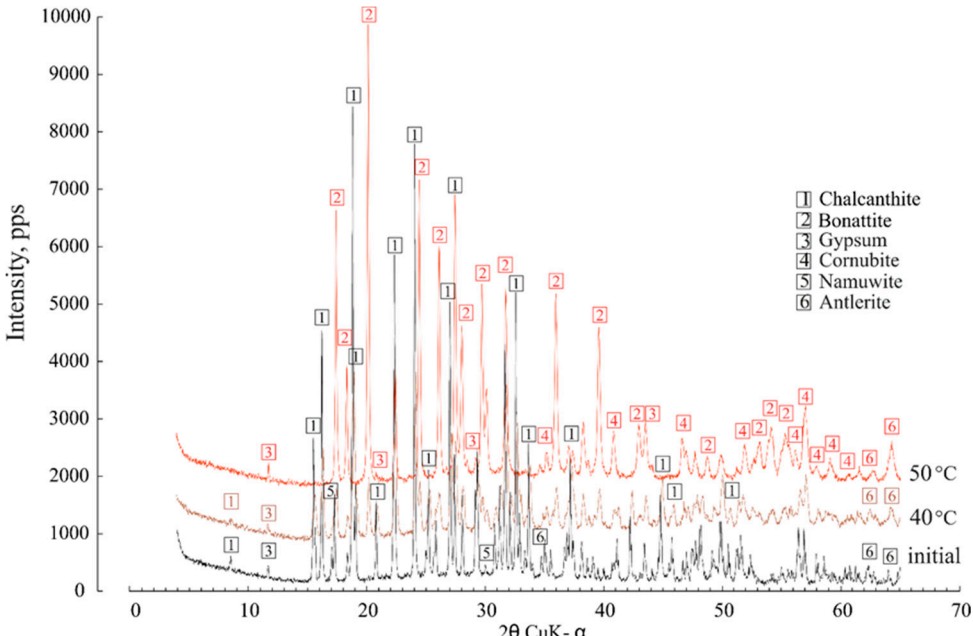

**Figure 7.** The effect of heating at 40 °C and 50 °C on the chalcanthite XRD characteristics.

Thus, it was established that the melanterite and chalcanthite lost their adsorbed and structural water when heated, while only sorbed water was released from the Na-jarosite surface.

## 4. Discussion

Sulfate salts are abundant in deposits and mining environments [1–9]. Their numbers extend well beyond the simple sulfate salts that have thus far been investigated experimentally, as described in [17]. Most experimental studies have been limited to simple salts with metals in a single valence state, combined with sulfate and various numbers of water molecules. Examples include the $Fe^{2+}SO_4$–$H_2O$, $Cu^{2+}SO_4$–$H_2O$, and $MgSO_4$–$H_2O$ systems [31], and $ZnSO_4$–$H_2O$ [32]. Sulfate minerals are important minor constituents of high-temperature hydrothermal systems and waste heaps, such as the burning Belovo waste. Simple metal-sulfate salts are rare in high-temperature technogenic settings due to the complexity of pyrometallurgical enrichment material [14,25]. Nevertheless, a comparison with stoichiometric minerals forms a basis for discussion. The diagram from [17], showing the location of dehydration reactions in simple sulfate systems is presented in Figure 8. The minerals that were not found in our study are listed in Table S1 for convenience. The gray field marks the range of temperature/humidity conditions for the laboratory experiments in this article (25–60 °C), according to [33]. Jarosite is not shown because we considered only simple reactions, where the only chemical difference among the minerals was the number of hydration water molecules in the mineral formula; for example, melanterite-rozenite (2):

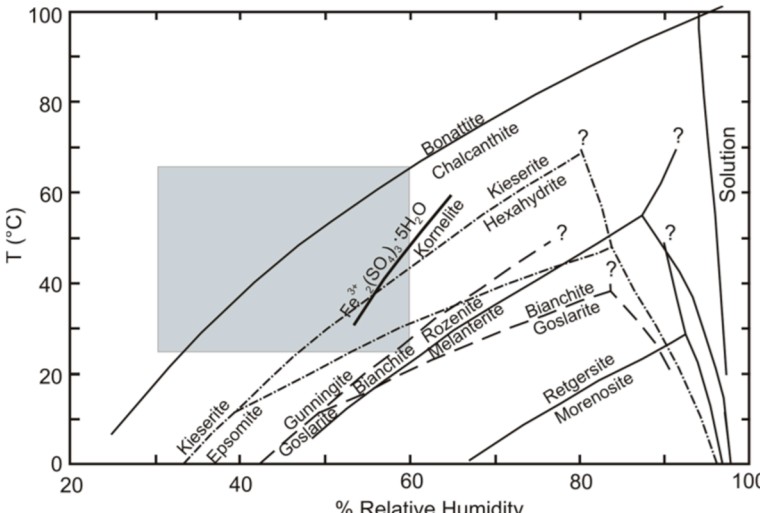

**Figure 8.** Diagram showing the location of dehydration reactions, in terms of temperature and relative humidity, for the simple sulfate systems that were investigated experimentally, using the humidity-buffer technique [17]. The gray field shows the range of temperature conditions for the laboratory experiments in this article (25–60 °C). The humidity values correspond to [33].

$$Fe^{2+}SO_4 \cdot 7H_2O = Fe^{2+}SO_4 \cdot 4H_2O + 3H_2O. \tag{2}$$

melanterite       rozenite

Melanterite and rozenite are the most common ferrous sulfates found at most metallurgical waste sites; the dehydration reaction relating these two phases lies in the lower right corner of the range of our experimental conditions (Figure 8). This means that melanterite dehydration occurred at the lowest temperatures. Since stable admixtures of Cu and Zn are present in the melanterite composition (0.7 wt%), it actually transforms to siderotil (3):

$$(Fe,Cu)SO_4 \cdot 7H_2O = (Fe,Cu)SO_4 \cdot 5H_2O + 2H_2O. \tag{3}$$

Cu-melanterite       Cu-siderotil

The dehydration reaction relating chalcanthite and bonattite lies in the upper left corner of the range of our experimental conditions (Figure 3). This represents the highest temperatures of chalcanthite dehydration, corresponding to our data. With respect to other contaminant elements, the following tentative range can be built: Retgerzite (Ni) → goslarite-nianchite (Zn) → melanterite-rozenite (FeII) → bianchite-gunningite (Zn,Mn,Fe) → epsomite-hexahydrite → hexahydrite-hieserite → chalcanthite-bonattite. Herein, the dominant cupric sulfate is chalcanthite, and bonattite is a rare mineral.

The considered range could explain, to some extent, the ambiguous dynamics of the separation of the elements in the vapor phase. The general scenario is a decrease in the concentrations of Ca, Mg, Sr, Na, K, Si, and Al with increasing temperature (Table 3, Figure 9). These elements are connected with the adsorbed water and atmospheric $CO_2$ held on the surfaces of the material by electrochemical forces. Therefore, the first-released portion formed over a long period of time, and so contains a greater quantity of elements. The second and following parts of $H_2O$ and $CO_2$ were adsorbed in the laboratory.

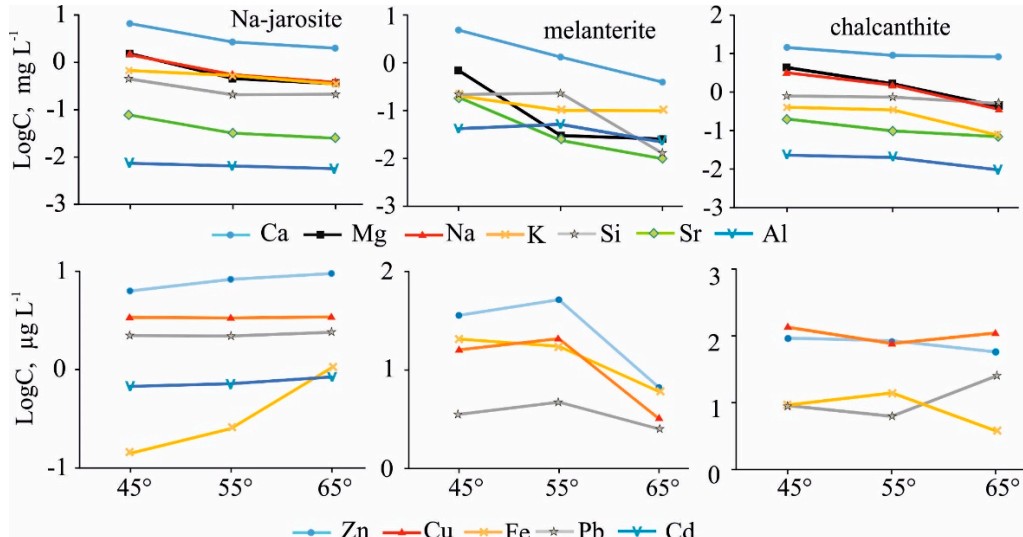

**Figure 9.** Dynamics of element migration with the vapor-gas phase, at different temperatures, from the three main minerals studied.

In the Na-jarosite gas condensates, the content of heavy metals (Cu, Zn, Pb, Co, Ni), and especially Fe, increased. We assume that was caused by temperature-accelerated dissolution of the grain surfaces by the heated sorbed water. In the experiment with melanterite, the concentrations of the metals increased noticeably in the gas condensates of the second stage (55 °C), when the melanterite was completely destroyed through the volatilization of adsorbed and structural (chemically-combined) waters. The concentration of Fe in the condensates, from resistant Na-jarosite, was two orders lower than in the melanterite condensates at 45 °C. In the third stage (65 °C), the Fe concentration in the condensates differed by a factor of only six. The content of Cu (130 μg/L) and Zn (93 μg/L) was highest in the chalcanthite gas condensate at 45 °C (when chalcanthite lost two molecules of water and transformed to bonattite). Namuwite and goslarite were the sources of the Zn in these condensates.

The nonlinear change in the elemental concentrations in the gas condensates at different temperatures reflects the complex compositions of the samples—the presence of a variety of crystalline hydrates with different impurities.

The activity of bacterial strains (Microbacteriaceae, Flavobacteriaceae, Moraxellaceae, Micrococcaceae, and Bacillaceae) can lead to the appearance of organic matter in vapor-gas phases. The formation of acetates and formates is possible primarily a result of uncultivated bacteria belonging to the primary anaerobes forming unfermentable products that include acetate and formate. Only a relatively small number of aerobes—for example, representatives of the genus, *Bacillus* [34]—are able to form acetate as metabolites under anoxic conditions. Usually, these bacteria live together with acetotrophs, using lactate and formate as sources of energy and carbon, with an external oxidizer ($Fe^{3+}$, $O_2$). Acetotrophs include a variety of nonculturable bacteria, particularly methanogens, as well as the representatives that were cultivated in the analyzed samples, including the families, Micrococcaceae and Microbacteriaceae, and particularly *Micrococcus luteus* [35], which was one of the most frequently recorded species in the analyzed samples.

## 5. Conclusions

Various elements' volatilization and vapor phase transport were proven in laboratory experiments during stepwise heating of sulfate minerals to a temperature of 65 °C. The highest TDS (mineralization) was found in the condensates from melanterite (59 mg/L due to the high content of $Cl^-$ and $NO_3^-$), followed by chalcanthite (29 mg/L) and Na-jarosite (17 mg/L).

The highest TDS was determined during the first sample heating at 45 °C, where contents of the common sulfate ion were: 9.3, 5.6, and 0.53 mg/L, respectively. An overall correlation between the

composition of the solid phases and gas condensates was absent due to the impurity minerals and non-stoichiometric composition of technogenic sulphates.

Major and trace elements in the condensate can be trapped by water vapor and can migrate with the vapor phase during the desorption and dehydration of hydrous sulfates. The content of the major cations (Ca, Mg, Na), Si, and some metals (Sr, Cu, Zn, Pb) was much higher in the condensates separated from chalcanthite. The concentrations of Fe, Al, Ba, Mn, and P were higher in the condensates from melanterite.

Dehydration of melanterite began at 30 °C, chalcanthite at 40 °C, and Na-jarosite remained stable in the temperature range up to 65 °C. Structural water was separated from melanterite and chalcanthite and new minerals (siderotil from melanterite and bonattite from chalcanthite) formed.

The following families of bacteria were cultivated: Microbacteriaceae, Flavobacteriaceae (*Flavobacterium* sp.), Moraxellaceae (*Acinetobacter* sp.), Micrococcaceae, and Bacillaceae (*Bacillus* sp.) in the efflorescence samples. The acetates and formates in the vapor-gas phases were due to the activity of strains belonging to the primary anaerobic families of these bacteria. The formation of volatile organometallic complexes could promote the volatilization of metals and their mobility in the gas phase.

**Supplementary Materials:** The following are available online at http://www.mdpi.com/2075-163X/9/6/344/s1, Figure S1: Scheme of the laboratory experiments for condensate collection, Table S1: Mineral compositions of the efflorescences.

**Author Contributions:** Conceptualization, S.B.; methodology, N.Y., S.B., N.A., A.D.; validation, S.B.; formal analysis, S.B.; investigation, S.B., N.A., O.S. (Olga Saeva), O.S. (Olga Shuvaeva), N.P., N.Y., T.K., V.Z., V.C., A.R.; resources, V.Z.; writing—original draft preparation, S.B.; writing—review and editing, S.B., O.G., N.Y., N.A.; visualization, S.B., N.A.; supervision, S.B.; project administration, S.B.; funding acquisition, S.B.

**Funding:** This research was funded by the RUSSIAN FOUNDATION FOR BASIC RESEARCH, grant number 17-05-00056 (mineralogical studies and condensate analyses), RUSSIAN SCIENCE FOUNDATION, grant number 19-17-00134 (microbiological section) and a BASIC RESEARCH PROJECT number 0331-2019-0012.

**Acknowledgments:** The authors gratefully thank Y.V. Seretkin for the helpful comments and recommendations on this manuscript. We are grateful to the anonymous reviewers for the valuable suggestions, as well as to the Assistant Editors Francis Wu and Lucas Xiang for the editorial handling of the manuscript.

**Conflicts of Interest:** The authors declare no conflict of interest. The funders had no role in the design of the study; in the collection, analyses, or interpretation of data; in the writing of the manuscript; or in the decision to publish the results.

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
