# Peer review of "Gas Transfer of Metals during the Destruction of Efflorescent Sulfates from the Belovo Plant Sulfide Slag, Russia"

_minerals, doi:10.3390/min9060344_

Round 1
Reviewer 1 Report
The manuscript entitled “Gas transfer of metals during the destruction of efflorescent sulfates from the Belovo plant sulfide slag, Russia” by Bortnikova et al., presents an interesting study of sulfate efflorescence samples, with particular focus on the chemistry of vaporous phase resulted from their volatilization. This kind of approach may be of great interest for AMD studies in different regions around the globe, and helps to a better understanding of AMD products and their behavior under specific conditions.
The manuscript is well-written, having a clear structure and in my opinion, with a good English that makes it easy to read. The methods, results and interpretation sections are concise and clearly presented. The paper is of reasonable length and the figures and tables are good and necessary.
The methods are described in detail, and the samples are very well characterized as regarding their mineralogy with different sulfate phases, transformation processes and composition of gas phases. The discussions and conclusions are supported by the results.
Therefore, I recommend the acceptance of this manuscript in the Minerals journal. Yet, the authors should address some minor suggestions, listed below.
Specific comments:
- the mineral names should be written under the chemical reaction formulas (equations 1-3), for individual phases involved into reaction (lines 74, 264, 269);
- Results, section 3.1. – I recommend to write also the chemical formula for each sulfate mineral, at their first mention in text (for example: Na-jarosite – line 159; melanterite – line 159; chalcanthite – line 171, etc.);
- Table 3 – please explain in table or in a footnote the “Tm” and “Td” terms;
- the methods section presents also a microbiological study, but in the Results there is no reference to it, only a small discussion. Maybe some results of the bacterial experiment should be interesting.

Author Response
Dear colleague, thank you very much for your time spent reading the manuscript and for the kind and valuable suggestions. We have tried to accommodate all your comments.
Replies to specific comments:
the mineral names should be written under the chemical reaction formulas (equations 1-3), for individual phases involved into reaction (lines 74, 264, 269);
The mineral names have been added.
Results, section 3.1. – I recommend to write also the chemical formula for each sulfate mineral, at their first mention in text (for example: Na-jarosite – line 159; melanterite – line 159; chalcanthite – line 171, etc.);
The chemical formulas have been written.
Table 3 – please explain in table or in a footnote the “Tm” and “Td” terms;
Explanation of Tm and Td have been added in the text.
the methods section presents also a microbiological study, but in the Results there is no reference to it, only a small discussion. Maybe some results of the bacterial experiment should be interesting.
We have added some results of the bacterial experiment in the section Results.
Reviewer 2 Report
This is a well-written paper describing sound experimental work. I have two concerns with the presentation of the study and no concerns with the science and methodology behind the work.
First, the introduction presents good background information, but I found it lacking in a clear description of the purpose of the research and the anticipated results and potential meaning. The acid formation is described as being a critically interesting aspect of these phase transitions of sulfate-bearing salts, but little time is spent foreshadowing the possible results and their potential meanings to the study and the literature as a whole. Also, the introduction discusses the possibility, and importance, of creating a predictive model; however, this is not discussed any further in the paper.
Second, the conclusion is short and does not tie together all of the open questions raised by the introduction section. The findings regarding the influence of the microbiological process are interesting and may deserve more discussion in the conclusion. The introduction begins by saying that "accurate classification of the acid-forming potential of waste rock is vital", but the conclusion falls short of describing how these findings support this process and add to our understanding and predictability of these acid-forming processes.
Author Response
Dear colleague, first of all, many thanks for your work under manuscript and very useful suggestions. We made changes to the text according to your comments.
First, the introduction presents good background information, but I found it lacking in a clear description of the purpose of the research and the anticipated results and potential meaning.
Thank you, description of the purpose and potential meaning have been added.
The acid formation is described as being a critically interesting aspect of these phase transitions of sulfate-bearing salts, but little time is spent foreshadowing the possible results and their potential meanings to the study and the literature as a whole. Also, the introduction discusses the possibility, and importance, of creating a predictive model; however, this is not discussed any further in the paper.
Yes, this sentence regarding a predictive model has been deleted.
Second, the conclusion is short and does not tie together all of the open questions raised by the introduction section. The findings regarding the influence of the microbiological process are interesting and may deserve more discussion in the conclusion. The introduction begins by saying that "accurate classification of the acid-forming potential of waste rock is vital", but the conclusion falls short of describing how these findings support this process and add to our understanding and predictability of these acid-forming processes.
We have added some result of the bacterial experiment in the conclusion.
Yes, you are right, the acid-forming potential is not the purpose of this work. We deleted this sentence from the Introduction section.
Reviewer 3 Report
This paper describes some of the geochemical features of a sulphidic slag pile, particularly with respect to the secondary minerals and their chemical signatures.
The paper is well written and interesting, however, I feel the context of the paper could be championed, for example, what is the global significance of the findings- how does this help to improve the management of similar piles going forward? Further, how does the element cycling controlled by these secondary phases and gases relate to the broader environment of the study site? Without this context, the paper reads like a very interesting study which gives information on element cycling and not much more. See my comments on the returned manuscript for more detailed edits.

Author Response
Dear Reviewer,
Please, find attached the pdf file with our answers and corrections to your helpful comments.
